# Modular assembly of dynamic models in systems biology

**Michael Pan** [1,2,3]*, **Peter J. Gawthrop** [1], **Joseph Cursons** [4], **Edmund J. Crampin** [1,2,3,5]†

**1** Systems Biology Laboratory, School of Mathematics and Statistics, and Department of Biomedical Engineering, University of Melbourne, Parkville, Victoria, Australia, **2** ARC Centre of Excellence in Convergent Bio-Nano Science and Technology, Faculty of Engineering and Information Technology, University of Melbourne, Parkville, Victoria, Australia, **3** School of Mathematics and Statistics, Faculty of Science, University of Melbourne, Parkville, Victoria, Australia, **4** Department of Biochemistry and Molecular Biology, Monash Biomedicine Discovery Institute, Monash University, Melbourne, Victoria, Australia, **5** School of Medicine, University of Melbourne, Parkville, Victoria, Australia

† Deceased.
* pan.m@unimelb.edu.au

**Data Availability Statement:** The code for this manuscript is available at https://github.com/mic-pan/Modularity-SysBio.

## Abstract

It is widely acknowledged that the construction of large-scale dynamic models in systems biology requires complex modelling problems to be broken up into more manageable pieces. To this end, both modelling and software frameworks are required to enable modular modelling. While there has been consistent progress in the development of software tools to enhance model reusability, there has been a relative lack of consideration for how underlying biophysical principles can be applied to this space. Bond graphs combine the aspects of both modularity and physics-based modelling. In this paper, we argue that bond graphs are compatible with recent developments in modularity and abstraction in systems biology, and are thus a desirable framework for constructing large-scale models. We use two examples to illustrate the utility of bond graphs in this context: a model of a mitogen-activated protein kinase (MAPK) cascade to illustrate the reusability of modules and a model of glycolysis to illustrate the ability to modify the model granularity.

## Author summary

The biochemistry within a cell is complex, being composed of numerous biomolecules and reactions. In order to develop fully detailed mathematical models of cells, smaller sub-models need to be constructed and connected together. Software and standards can assist in this endeavour, but challenges remain in ensuring that submodels are both consistent with each other and consistent with the fundamental conservation laws of physics. In this paper, we propose a new approach using bond graphs from engineering. In this approach, connections between models are defined using physical conservation laws. We show that this approach is compatible with current software approaches in the field, and can therefore be readily used to incorporate physical consistency into existing model integration methodologies. We illustrate the utility of this approach in streamlining the development of models for a signalling network (the MAPK cascade) and a metabolic network (the

**Funding:** This research was in part conducted and funded by the Australian Research Council Centre of Excellence in Convergent Bio-Nano Science and Technology (project number CE140100036, http://purl.org/au-research/grants/arc/CE140100036), awarded to EJC. MP was supported by a Postdoctoral Research Fellowship from the School of Mathematics and Statistics, University of Melbourne. PJG was supported by the Faculty of Engineering and Information Technology, University of Melbourne via a Professorial Fellowship. The funders had no role in study design, data collection and analysis, decision to publish, or preparation of the manuscript.

**Competing interests:** The authors have declared that no competing interests exist. Author Edmund Crampin was unable to confirm their authorship contributions. On their behalf, the corresponding author has reported their contributions to the best of their knowledge.

glycolysis pathway). The advantage of this approach is that models can be developed in a scalable manner while also ensuring consistency with the laws of physics, enhancing the range of data available to train models. This approach can be used to quickly construct detailed and accurate models of cells, facilitating future advances in biotechnology and personalised medicine.

## Introduction

Over the past few decades, advances in both data generation and computational resources have enabled the construction of large-scale kinetic models in systems biology, including whole-cell models that represent every known biomolecule in the cell [1]. An accurate and robust whole-cell model can provide several benefits to the community: data on specific organisms can be cross-evaluated and reconciled [2]; simulations could be used to rule out fruitless experiments and clinical trials; the models themselves could be used as a basis for designing novel circuits in synthetic biology; and fundamental questions about biology may be addressed in a holistic and systematic manner [3, 4].

The first comprehensive whole-cell model was developed for *Mycoplasma genitalium* [1] and there are ongoing efforts to develop whole-cell models of *Escherichia coli* [5] and human cells [6]. However, it has been acknowledged that the highly manual practices used in the development of the initial model of *M. genitalium* are unlikely to scale up to more complex organisms. The biomodelling community has identified several potential roadblocks to whole-cell modelling, including the lack of sufficient biological knowledge and data, model incompatibility, inadequate model development tools, inadequate model formats and parameter uncertainty [4, 6].

This paper addresses the issue of approaching model development in a modular manner. Typical requirements for such model development strategies involve reusing and integrating submodels together into more comprehensive models, and swapping between alternative models of the same system for benchmarking and comparison [7–9]. There have been several software-related developments in the systems biology community focussing on improving the reusability of models, some of which are beginning to be used in whole-cell modelling [10]. However, ensuring the reusability of fully integrated cell models remains a challenge [11]. While adequate software frameworks are essential to the modular development of large-scale kinetic models, an understanding of the physics of biological systems is also necessary to address issues in model compatibility and provenance. In particular, the laws of thermodynamics have been invoked in the context of metabolic modelling, allowing modellers to estimate the energetic favourability of reactions [12], to constrain fluxes in constraint-based models [13, 14] and to improve parameter estimation in large-scale kinetic models [15–17] and whole-cell models [5]. This paper argues that the concepts of module interconnection from physics and thermodynamics are consistent with current model development practices in systems biology, and we suggest the use of bond graphs (from the discipline of engineering) as a framework for unifying developments from both software development and biological thermodynamics.

We begin by defining modularity in systems biology and arguing that an improved understanding of biophysics can contribute to this area. We then use biochemical examples to illustrate how bond graphs incorporate physical constraints into a modular framework for systems biology. In the Results section, we illustrate the benefits of this approach by applying the principles of modular development to bond graph models of a mitogen-activated protein kinase

(MAPK) cascade and glycolysis. Finally, we summarise ongoing developments in unifying modularity and thermodynamics in systems biology and conclude with some suggestions to enable the development of fully detailed models of cells.

## Methods

### Modularity in systems biology

Due to their complexity, large-scale models in systems biology need to be constructed by dividing the problem into manageable submodels. Early notions of modularity in biomodelling were borrowed from principles in engineering and software development [18, 19]. In those disciplines, modules can be defined as parts of a system that (a) retain their own identity and are often developed and operated independently, but interact with other parts of the system and (b) hide the details of their implementation from the rest of the system, except through pre-defined interfaces [9]. Using this notion of modularity, the parts of a module that are available for connection and communication are said to be "exposed".

However, the definition above—which is also known as "black-box" modularity—is not conducive to the incremental accumulation of knowledge that occurs in biology. Advancements in our understanding of biology may force modellers to interface with previously hidden components within existing models [7, 9]. It is becoming increasingly apparent that modules in biological modelling need to be more flexible than engineering modules. As a result, the notion of modularity in systems biology is far less clear than in the established disciplines of engineering and software development. In recent years, systems biology has favoured the use of a "white-box" approach to modularity in which modules do not completely hide the details of their implementation, but instead allow individual variables and components to be exposed as required [20].

Broadly speaking, notions of modularity used in systems biology can be categorised into *computational modularity*, the ability for models to communicate and interact with each other in a physically consistent manner; and *functional* (or *behavioural*) *modularity*, the ability of modules to be isolated from the effects of other modules. This paper will focus on computational modularity. The role of functional modularity is pivotal to systems and synthetic biology, particularly in reducing loading effects between engineered genetic circuits [21]. However, functional modularity can only be analysed and designed through the lens of computational modularity [22].

If handled correctly, modular model development can provide a number of benefits to modellers (see Fig 1), including:

1. Enabling large-scale models to be built from smaller submodels that communicate through clear and unambiguous interfaces.

2. Providing a framework for models to be developed, tested and validated in isolation before incorporating them into larger models.

3. Separating the description of model equations from the software implementation of the model (including simulation).

4. Allowing incremental changes to be made to existing models in light of new measurements or knowledge, and allowing the provenance of models to be tracked.

5. Enabling the abstraction of important modules, providing the means to instantiate multiple copies of repeated motifs and swapping out a submodel for another model with a different level of granularity.

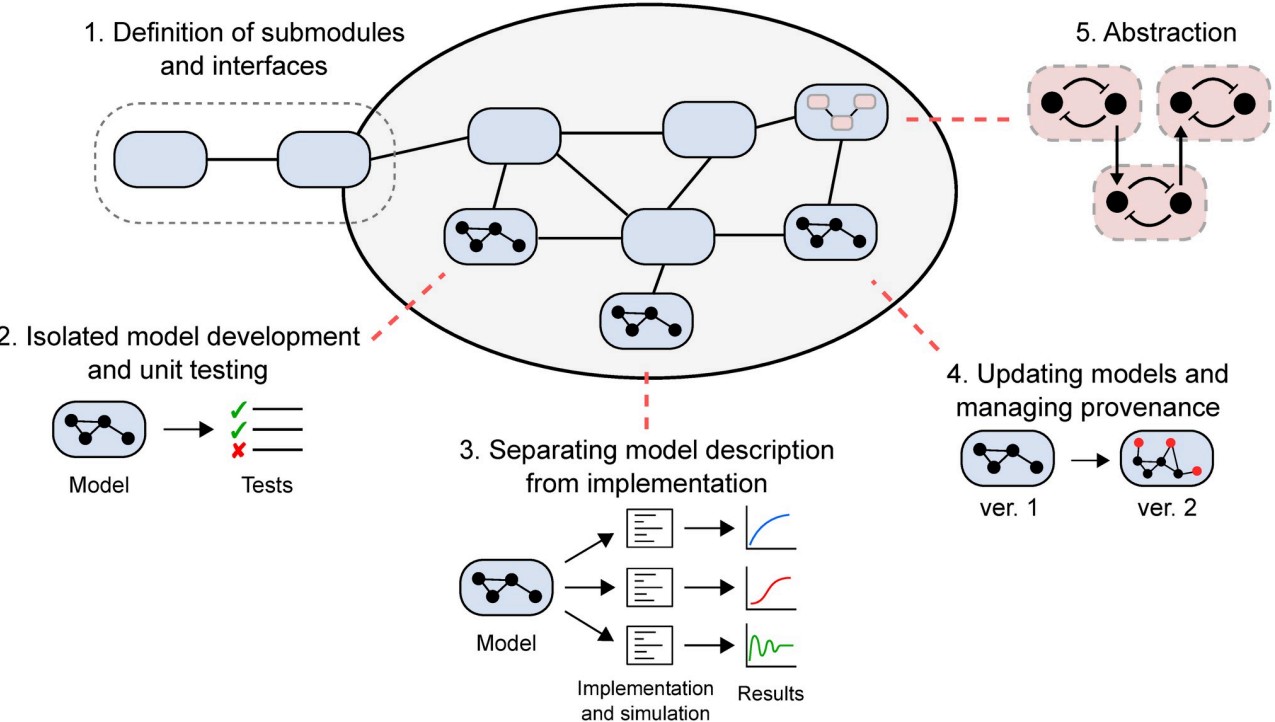

**Fig 1. The importance of modularity in models for systems biology.** Modularity can facilitate the construction of whole-cell models by (1) providing unambiguous and flexible interfaces for submodels to communicate; (2) allowing model development and unit testing to be done on individual submodels; (3) separating the description of the model from its implementation; (4) allowing models to be iteratively updated with a record of how the equations and parameters were derived; and (5) allowing repeated motifs to be abstracted into reusable structures.

Thus, modularity can facilitate collaborative efforts to build whole-cell models, allows models to be updated where necessary and enhances the usefulness of models beyond their initial publication.

Current approaches to model reuse and integration in systems biology can be broadly categorised into three approaches, ordered by increasing flexibility:

1. **Standard model description formats**. To enable the reuse of biomodels by different research groups, biomodellers have developed standards for describing models. Of these, the Systems Biology Markup Language (SBML) [23] and CellML [19] are two prominent examples. Once encoded within such standard model descriptions, analysis and simulation can be run on separately developed software such as OpenCOR (CellML) and COPASI (SBML) [24, 25]. The simulation protocols can themselves be specified using the Simulation Experiment Description Markup Language (SED-ML) [26].

2. **Biological modularity**. Biological-level modularity introduces white-box modularity to systems biology by annotating model variables and parameters with standardised, machine-readable ontological terms [9]. This enables a strategy where software can automatically compose separately developed models together [9, 20]. SemGen and semanticSBML are two software tools that implement biological-level modularity [20, 27]. SBML also supports both white and black box modularity through its hierarchical package [28], which is implemented in the graphical tool iBioSim [29].

3. **Programmatic approaches**. The "programmatic approach" to modelling was developed to allow models to integrate together in a flexible manner, but also to address the relative inflexibility of standard modelling languages. In the programmatic approach, models are treated as declarative programming objects rather than mathematical equations [7]. Using this approach, models can be embedded within programming languages such as Python. This approach automates model construction by allowing models to be defined at different hierarchical levels, for example by generating equations through the specification of macros for repeating motifs. Two implementations of the programmatic approach are *little b* [7] and PySB [30]. More recently, the BondGraphTools package has been developed to introduce thermodynamics into such an approach [31].

Note that these approaches are not independent of each other, and many modelling frameworks use several of these approaches [32, 33].

Despite recent computational advances in enabling the modular development of biomodels, there remain key limitations in current approaches:

1. There is no guarantee that the integrated model will be consistent with basic physical principles such as conservation of mass, charge and energy.

2. It remains difficult to resolve points of conflict between models, such as conflicts between parameters and assumptions.

3. There is limited scope for dealing with multi-physics systems that arise in electrophysiology and mechanochemistry.

Resolving these issues requires the conservation laws of physics to be embedded within computational modules. Network Thermodynamics, using bond graphs, is a modelling framework that fits with the requirement of developing physically consistent models, while retaining compatibility with existing approaches.

## Bond graphs

Bond graphs provide a modular framework for constructing physically and thermodynamically consistent models in systems biology. The framework was first applied to biology by Oster, Perelson and Katchalsky in the context of Network Thermodynamics, as a method for incorporating the laws of thermodynamics into theoretical models of living systems [34, 35]. This work followed in the tradition in physics and engineering that if you "get the physics right", "the rest is mathematics" [36, 37]. Bond graph models are defined by combining constitutive relations with physical conservation laws, giving rise to a declarative model structure. This confers some advantages from a modelling perspective:

1. Models can be specified in terms of physical connections between components, giving rise to a graphical representation of the model equations, which are consistent with the conservation laws of physics.

2. Bond graphs inherently support modular modelling, as components can easily be swapped in and out without affecting the high-level model structure.

3. Due to the fundamental nature of energy in all physical systems, a thermodynamic approach can be used to link together models of systems from different physical domains such as the electrical, mechanical, chemical and hydraulic domains. Therefore, bond graphs models can be constructed for a wide range of multi-physical biological systems, including electrophysiology and mechanobiology [38–40].

There has been a long history of thermodynamic modelling for biochemical reaction networks [5, 13, 15]. In this section, we introduce bond graphs as an intuitive method for embedding such approaches within a modular framework.

**An explicit graphical representation of biochemical systems.** We first use a simple example to illustrate the how the structure of a bond graph encodes differential equations [35, 41, 42]. Consider the enzyme-catalysed reaction in Fig 2A, noting that all chemical reactions are thermodynamically reversible. Assuming that the reactions follow the law of mass action, the system can be described using the differential equations

$$\frac{dx_E}{dt} = -v_1 + v_2 \tag{1a}$$

$$\frac{dx_C}{dt} = v_1 - v_2 \tag{1b}$$

$$\frac{dx_S}{dt} = -v_1 \tag{1c}$$

$$\frac{dx_P}{dt} = v_2 \tag{1d}$$

where the fluxes through the reactions $v_1$ and $v_2$ are given by

$$v_1 = k_1^+ x_E x_S - k_1^- x_C \tag{2a}$$

$$v_2 = k_2^+ x_C - k_2^- x_E x_P. \tag{2b}$$

In the above equations, the rate parameters $k_1^+$, $k_1^-$, $k_2^+$ and $k_2^-$ are defined in Fig 2A, and $x_E$, $x_C$, $x_S$, $x_P$ are the concentrations (or molar amounts) of E, C, S and P respectively.

Because it is often useful to consider rate laws independently from the stoichiometry of the system, systems biologists may favour an expanded representation of the network as shown in Fig 2B, where the reactions and species reside in their own components. This representation follows the Systems Biology Graphical Notation (SBGN) standard [43].

The bond graph representation in Fig 2C is a further expansion of the diagram in Fig 2B. This representation firstly adds two physical variables to the edges: a chemical potential $\mu$ [J/mol] (blue variables) and molar flux $v$ [mol/s] (green variables). Since $\mu$ and $v$ multiply to give power $P$ [J/s], each connection transfers energy between components. In addition, separate nodes ($\bullet$ and $\blacktriangledown$) are used to model mass and energy conservation laws inherent within these systems, discussed further below.

Every component (node) within the system contains its own independent set of equations and parameters. Each chemical species (open circles $\bigcirc$ in Fig 2C) is associated with a chemical potential $\mu$. In dilute systems at constant temperature and pressure, this quantity is related to abundance $x$ by

$$\mu = RT \ln(Kx) \tag{3}$$

where $x$ [mol] is the amount of the species, $K$ [mol$^{-1}$] is the thermodynamic parameter for that species, $R = 8.314$ JK$^{-1}$mol$^{-1}$ is the ideal gas constant and $T$ [K] is the absolute temperature.

## A

### Chemical reactions

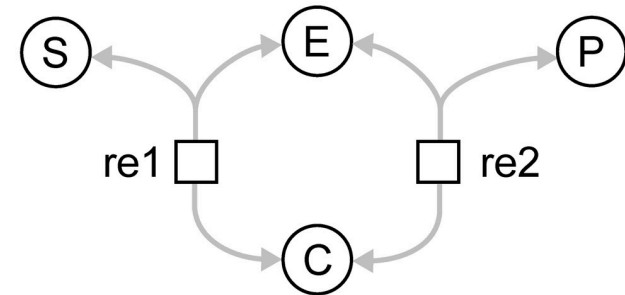

## B

### Graph representation

## C

### Network thermodynamics representation

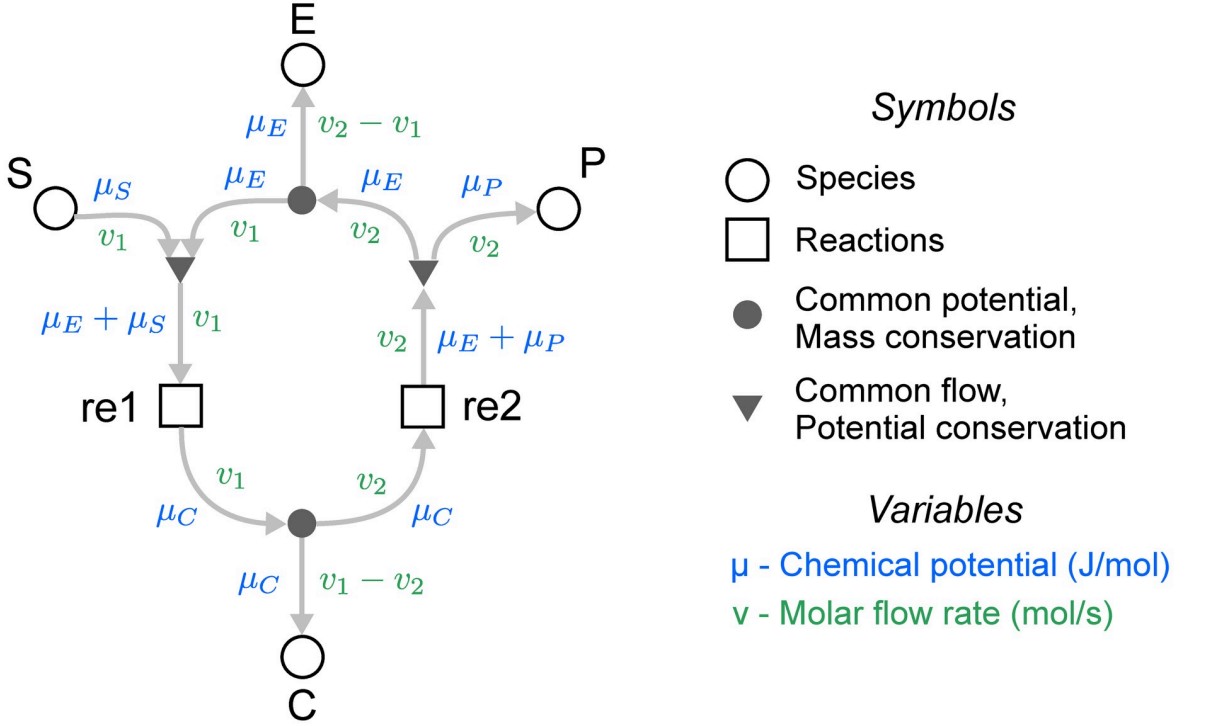

**Fig 2. The energetics of an enzyme-catalysed reaction. (A)** Chemical reaction scheme. **(B)** A graph representation of the reaction scheme typically seen in systems biology. Species are represented as circles and reactions are represented as squares. The grey arrows indicate the flow of mass. **(C)** A bond graph representation of the network. Note that in contrast to the graph representation in (B), additional elements have been added to the representation to represent conservation of mass (closed circles ●) and conservation of energy (triangles ▼). The arrows here represent the molar flow rate (green) and the associated chemical potential (blue), thus the flow of both mass and energy is accounted for. The arrowheads indicate the direction of positive flux, but all reactions can proceed in the reverse (negative) direction as well.

The parameter $K$ is related to the standard free energy of the species; one can also write Eq 3 as

$$\mu = \mu^0 + RT \ln(c/c^0) = \mu^0 + RT \ln\left(\frac{x}{c^0 V}\right) \tag{4}$$

where $c$ [M] is the concentration of species, $V$ [L] is the volume of the compartment and $\mu^0$ [J/mol] is the standard chemical potential taken at a concentration of $c^0$ (for simplicity, $c^0$ is often taken to be 1M). By equating Eqs 3 and 4, $K$ is related to $\mu^0$ through the equation

$$K = \frac{1}{c^0 V} \exp(\mu^0/RT). \tag{5}$$

Similarly, the rate of each reaction (squares □ in Fig 2C) is given by a constitutive relationship between reaction rate $v$ and the thermodynamic potentials. For example, the thermodynamic Marcelin-de Donder equation represents reversible mass action kinetics [42]:

$$v = \kappa\left[\exp\left(\frac{A^f}{RT}\right) - \exp\left(\frac{A^r}{RT}\right)\right] \tag{6}$$

where $A^f$ ($A^r$) is the forward (reverse) affinity, or the sum of chemical potentials within the reactants (products). We note that while we have used $K$ and $\kappa$ as our parameters, these values can also be expressed in terms of energetic quantities such as the free energy of formation (see Appendix A in S1 Text).

Therefore, the species in Fig 2C encode the relationships

$$\mu_E = RT \ln(K_E x_E) \tag{7a}$$

$$\mu_C = RT \ln(K_C x_C) \tag{7b}$$

$$\mu_S = RT \ln(K_S x_S) \tag{7c}$$

$$\mu_P = RT \ln(K_P x_P) \tag{7d}$$

and the reactions encode the relationships

$$v_1 = \kappa_1\left[\exp\left(\frac{A_1^f}{RT}\right) - \exp\left(\frac{A_1^r}{RT}\right)\right] \tag{8a}$$

$$v_2 = \kappa_2\left[\exp\left(\frac{A_2^f}{RT}\right) - \exp\left(\frac{A_2^r}{RT}\right)\right]. \tag{8b}$$

To obtain the correct fluxes for each reaction, the chemical potentials $\mu$ of the species need to be correctly mapped onto the reaction affinities $A^f$ and $A^r$. Because reactions 1 and 2 are connected directly to $\mu_C$ in Fig 2C, it is clear that

$$A_1^r = A_2^f = \mu_C. \tag{9}$$

However, conservation of chemical potential (energy per mole) needs to be considered when determining $A_1^f$ and $A_2^r$. These affinities are related to the species potentials through the

relationships

$$A_1^f = \mu_S + \mu_E \tag{10a}$$

$$A_2^r = \mu_P + \mu_E. \tag{10b}$$

The above energy conservation equations are encoded within triangular (▼) components (Fig 2C), which constrain the model such that the sum of potentials of the edges directed into the triangles is equal to those directed outwards. Note also that fluxes $v$ of the connected edges are equal as each reaction consumes reactants and produces products at the same rate. These common flow junctions are analogous to Kirchhoff's voltage law in electrical circuits.

Finally, the fluxes through the reactions are related back to the rates of change in species through the conservation of mass components represented by the closed circles (●) in Fig 2C. These components constrain the fluxes such that the sum of fluxes into the component is equal to the sum of fluxes out of the component. These encode the mass balance equations in Eq 1. The common potential junction is analogous to Kirchhoff's current law in electrical circuits.

Once Eqs 1, 7–10 are combined, one can derive the differential equations

$$\frac{dx_E}{dt} = -\kappa_1 K_E K_S x_E x_S + \kappa_1 K_C x_C + \kappa_2 K_C x_C - \kappa_2 K_E K_P x_E x_P \tag{11a}$$

$$\frac{dx_C}{dt} = \kappa_1 K_E K_S x_E x_S - \kappa_1 K_C x_C - \kappa_2 K_C x_C + \kappa_2 K_E K_P x_E x_P \tag{11b}$$

$$\frac{dx_S}{dt} = -\kappa_1 K_E K_S x_E x_S + \kappa_1 K_C x_C \tag{11c}$$

$$\frac{dx_P}{dt} = \kappa_2 K_C x_C - \kappa_2 K_E K_P x_E x_P. \tag{11d}$$

This thermodynamic formulation has the same form as the kinetic formulation (Eqs 1 and 2) with the parameters redefined as

$$k_1^+ = \kappa_1 K_E K_S \tag{12a}$$

$$k_1^- = \kappa_1 K_C \tag{12b}$$

$$k_2^+ = \kappa_2 K_C \tag{12c}$$

$$k_2^- = \kappa_2 K_E K_P. \tag{12d}$$

While the thermodynamic formulation contains more parameters (6) than the kinetic formulation (4), they overcome a limitation of kinetic parameters. Whereas kinetic parameters are not free to be independently specified and require detailed balance constraints to be thermodynamically consistent, thermodynamic parameters can be chosen independently. Systems biologists have previously used thermodynamic parameters to avoid thermodynamically inconsistent model behaviour [15, 44]. More recently, the approach has been suggested for whole-cell modelling as a method for resolving points of conflict between data [5].

Thus, a strength of bond graph models in this context is that the differential equations of a biological network can be directly derived from the network structure, which paves the way for the modular construction of such models.

*A remark on notation*: Traditionally, the bond graph representation uses a textual notation for components rather than the graphical notation used in this paper. Specifically, species (◯) are represented as **C** components, reactions (□) as **Re** components, common potentials (●) as **0**-junctions and common flows (▼) as **1**-junctions. Furthermore, bonds are depicted using half-arrows rather than full arrows. However, in light of recent efforts to "modernise" the bond graph representation [45], we have depicted components in shapes rather than letters to make the representation closer to conventions seen in systems biology.

**A modular representation for enzymes.**   In systems biology, there are often several plausible equations for modelling enzyme-catalysed reactions. A modular approach is desirable in allowing one enzyme model to be swapped out for another [46]. We illustrate this by considering a modular version of the enzyme-catalysed reaction of Fig 2. The system can be represented using the diagram in Fig 3A, where S and P are connected via a yet-to-be-defined module shown by the light blue box. This arbitrary module can then be substituted for any component describing a plausible reaction mechanism.

Enzyme-catalysed reactions can be described by rate laws (Fig 3B). The simplest of these is the law of mass action in Eq 6 (Fig 3B; white box). This can be substituted for more complex kinetics, for example, the reversible Michaelis-Menten equation (orange box)

$$v = \bar{\kappa} e_0 \frac{e^{\mu_S/RT} - e^{\mu_P/RT}}{1 + \dfrac{e^{\mu_S/RT}}{R_{b0}} + \dfrac{e^{\mu_P/RT}}{R_{b1}}} \tag{13}$$

with the parameters $\bar{\kappa}$ (rate constant [s$^{-1}$]), $R_{b0}$ (binding constant of the substrate [dimensionless]), $R_{b1}$ (binding constant of the product [dimensionless]) and $e_0$ (total amount of enzyme [mol]) [42]. Thus, a bond graph approach allows alternative rate laws to be easily swapped for one another while retaining thermodynamic consistency. It is worth noting that the Michaelis-Menten rate law can be derived as a simplification of a more complex mass action model [47, 48].

In some cases, the full dynamics of the enzymatic reaction need to be considered. The advantage of a modular representation is that groups of reactions can be encapsulated into a model component. The diagram in Fig 3A can be converted into a simple two-state mechanism by defining the light blue box as the network in the left panel of Fig 3C; this is the same system as seen in Fig 2. Alternatively, to consider the conversion of substrate-bound enzyme to product-bound enzyme, the module defined in the right panel of Fig 3C could be used.

As seen in the above examples, parts of a module can be exposed by leaving open one end of a connection, which imposes a boundary condition on the model, allowing it to be connected to an external component. This is analogous to leaving ports open in electrical circuits. This kind of modularity provides tools for managing model complexity: generic modules are easily replicated and reused for different reactions that use the same mechanism, and the internal details of complex enzymatic mechanisms can be hidden. We now illustrate these ideas through the modular development of a MAPK signalling cascade model, and then by considering the glycolytic metabolic pathway modelled using different reaction rate laws.

## Results

### Modular development of a model of the MAPK cascade

The MAPK cascades are a family of biochemical signalling pathways that regulate important biological processes including growth, proliferation, migration and differentiation [49]. These

## A Modular representation

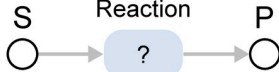

## B Generalised rate laws

## C Fully detailed kinetic mechanisms

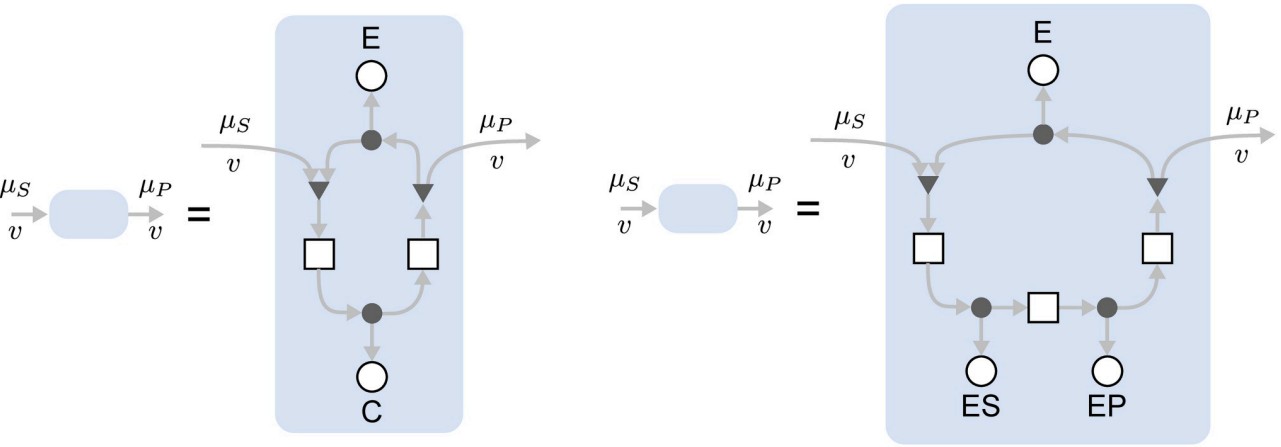

**Fig 3. Representation of enzymes as interchangeable modules.** **(A)** A modular representation of an enzyme. The module representing the enzyme, shown as a blue module, can be swapped depending on model requirements. (B) and (C) describe possible contents of the blue module. **(B)** Rate law representations of the enzyme, including mass action (white box) and the Michaelis-Menten equation (orange box). **(C)** Detailed multi-state representations of the enzyme, including the full two-state representation of the Michaelis-Menten enzyme (left) and a three-state representation with an explicit step for the conversion of substrate to product.

systems are composed of a series of phosphorylation events in which the phosphorylated substrate at one level of the cascade catalyses reactions at the next level, leading to signal amplification. From a modelling perspective, MAPK cascades contain a number of repeated motifs and therefore serve as interesting case studies for modular model development.

There are multiple MAPK cascades that naturally occur in eukaryotic cells. In this paper, we deal with the Mos/MAPK pathway found in *Xenopus* oocytes, a key regulator of maturation in these cells [50, 51]. MAPK cascades in human cells include the ERK-MAPK, c-Jun N-terminal kinase (JNK) and p38 MAPK pathways, which are of clinical relevance as they are implicated in both inflammation and cancer [52, 53]. While the kinetic properties may differ between pathways, a strength of taking a modular approach is that one can use similar network structures to account for many MAPK cascades.

**Core model.**   Here we construct a model of the Mos/MAPK cascade in a modular fashion using bond graphs. The core MAPK cascade model we considered was based on a study by Huang and Ferrell [50]. We chose this model in particular as it accounts for the elementary mass-action between enzyme states, which is essential when dealing with systems with coupled enzymatic reactions [54].

The presence of repeated network motifs consisting of kinases and phosphatases motivates an approach where generic modules corresponding to the kinases and phosphatases are first constructed, and then assembled into a model of the MAPK cascade. This gives rise to a model with a hierarchical structure. The modules for the kinase and phosphatase are defined in Fig 4A and 4B. These use mechanisms similar to the Michaelis-Menten model in the left panel of Fig 3C, but the free enzyme E, ATP, ADP and Pi need to be shared between modules and are therefore represented as external connections. X and XP are generic labels referring to the unphosphorylated and phosphorylated substrate respectively.

In the Methods section, it was clear how components were connected to the ports of each module. However, the increased number of ports in this example demands a more precise method of specifying external connections. For each module, each port is labelled in parentheses, i.e. *(label)*. These port labels are then used to define the connections to external components when the module is reused in a larger system—as indicated by the labels in red parentheses in Fig 4C and 4D. Using this notation, the kinase and phosphatase modules are combined into a generic model of a phosphorylation cycle (Fig 4C). Note that X and XP have been linked through a conservation of mass relationship, but the ports are otherwise exposed because all quantities are shared between modules in the full MAPK cascade model. In cases where there are multiple kinases and phosphatases operating in parallel, modules for these additional enzymes are easily added and connected to the mass conservation junction.

An advantage of defining generic modules is that multiple copies of these modules can be constructed and connected together. Since the MAPK cascade consists of multiple phosphorylation cycles, copies of the phosphorylation cycle module can be coupled together into a model of a full MAPK cascade (Fig 4D). Here, specific biomolecules are now assigned to the previously generic X and XP ports of the phosphorylation cycles. The multiple levels of the cascade are coupled by connecting the phosphorylated substrate in one level to the kinase port of the next cascade. We note that while this representation is abstracted, it is still a fully functional bond graph satisfying thermodynamic consistency. One could in principle flatten the bond graph by iteratively replacing modules with their definitions in Fig 4A–4C. Indeed, this approach can be used to algorithmically derive the equations of a bond graph model.

We chose model parameters to match the Huang and Ferrell [50] model as closely as possible. Because in that model the energetics of ATP hydrolysis were ignored and irreversible reactions were used, we reformulated the model to reintroduce both the effects of ATP hydrolysis and reversibility (details in Appendix A of S1 Text, with parameters given in S1 Table). Because the Huang and Ferrell model used irreversible reactions, an exact fit was impossible. Nonetheless, the reformulated model behaves almost identically to the original model (Figure A in S1 Text) under comparable physiological conditions.

We construct and run simulations of the model using the Python package BondGraphTools [31]; a tutorial for using this package can be found in Appendix C of S1 Text. The results are shown in Fig 5A, which plots the percentage of the activated kinase at each level of the cascade. We found that the concentrations settled to steady-state concentrations. These concentrations were recorded for different concentrations of input, resulting in the signal-response curves in Fig 5B. Under different input concentrations, the response of each substrate was sigmoidal. As expected from existing modelling studies [50], there was an amplification effect as the steepness of the transition from inactivated to activated forms became more pronounced towards

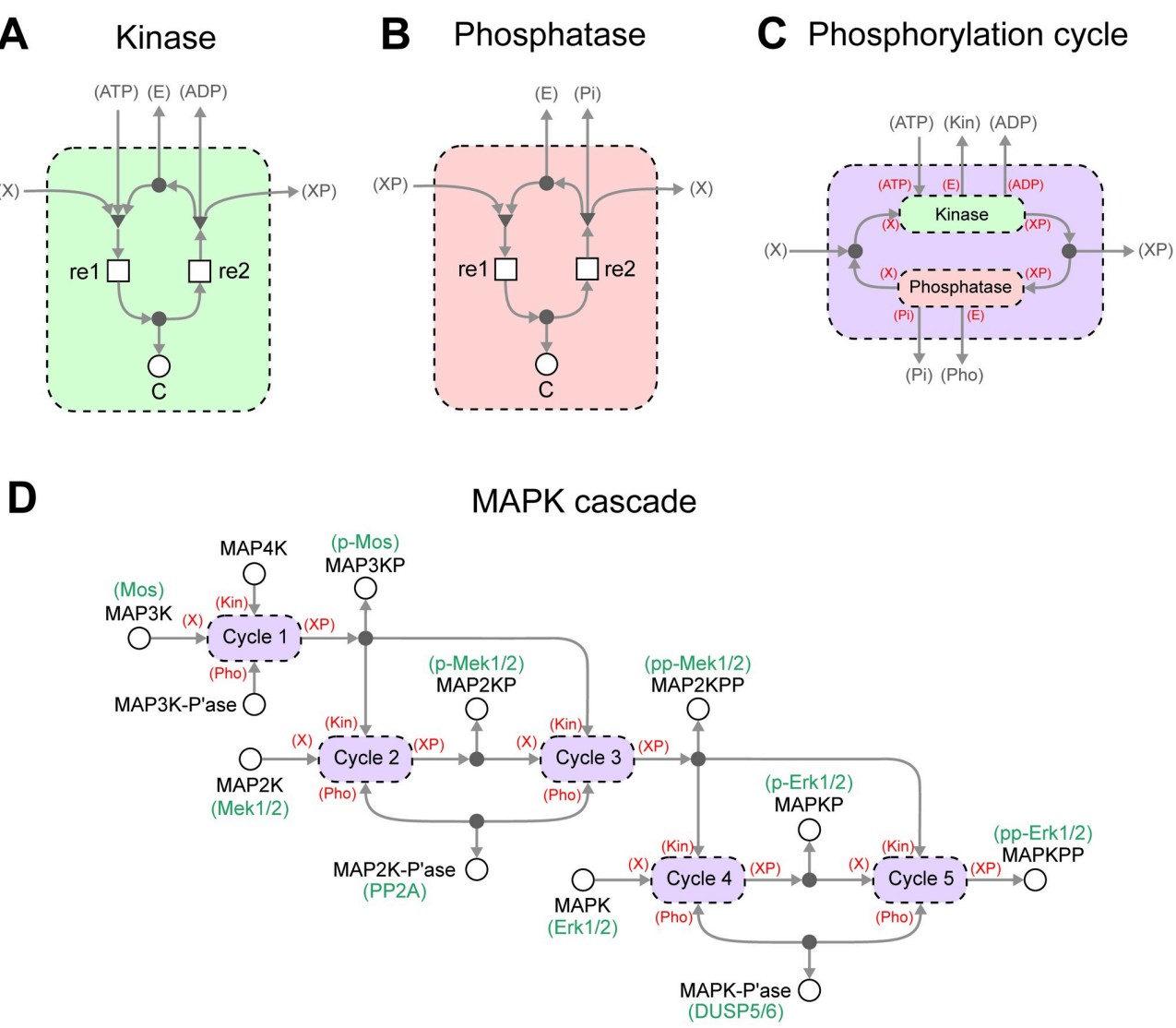

**Fig 4. A hierarchical and thermodynamic model of the MAPK cascade.** Generic modules can be made for **(A)** kinases and **(B)** phosphatases. **(C)** These modules can then be assembled into larger modules defining phosphorylation loops. To ease biological interpretation, the specific biological names are given in green where known. **(D)** Multiple copies of phosphorylation loops can be reused and connected to form a model of the MAPK cascade. For clarity, ATP, ADP and Pi have been omitted in (D).

the end of the cycle. When the energy supplied by ATP hydrolysis was reduced, a greater concentration of input was required to activate each of the kinases, with the effects being amplified at the lower levels of the cascade (Fig 5C).

While we took a black-box approach to modularity in this example, recent developments in bond graph modelling have enabled a more flexible white-box approach to modularity [55]. In Appendix B in S1 Text, we outline how this model could be constructed using such an approach where each module is itself a simulatable model.

**Incorporation of feedback.** An advantage of using a graphical and modular representation is that it is relatively easy to make incremental changes to models. We demonstrate this by modifying the model in Fig 4D—which we will now refer to as the "core" model—to include the effects of feedback. Both positive and negative feedback loops exist in MAPK cascades, but

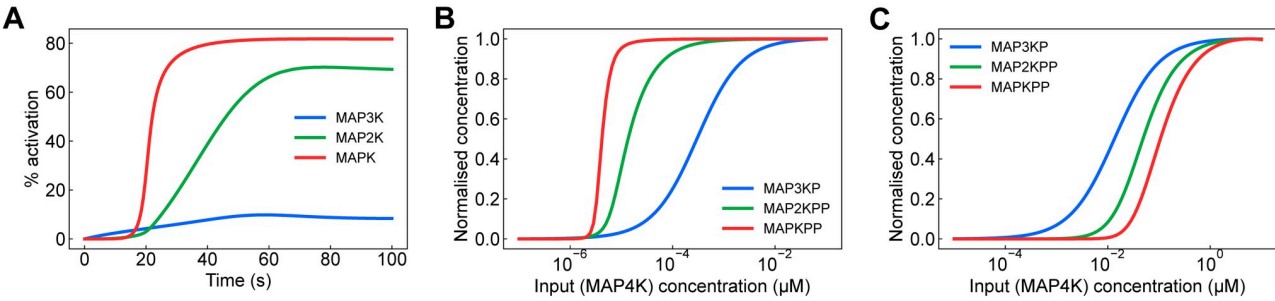

**Fig 5. Simulations of the model of the MAPK cascade. (A)** The activation of kinases over time. Activation is defined as the percentage of each kinase in its active state, i.e. the phosphorylated form of MAP3K and the biphosphorylated forms of MAP2K and MAPK. **(B)** The effect of input concentration on the steady-state concentrations of each of the activated kinases. Each curve is normalised to the highest concentration achieved for that species. **(C)** The activation curve in (B), but with reduced (80%) energy from ATP hydrolysis. The model was simulated with the initial conditions $x_{\text{MAP3K}} = 3$ nM, $x_{\text{MAP2K}} = 1.2$ $\mu$M, $x_{\text{MAPK}} = 1.2$ $\mu$M, $x_{\text{MAP3K-Pase}} = 0.3$ nM, $x_{\text{MAP2K-Pase}} = 0.3$ nM, $x_{\text{MAPK-Pase}} = 0.12$ $\mu$M. In (A), we initially set $x_{\text{MAP4K}} = 0.03$ nM, whereas this initial condition was varied for (B) and (C). All other species had an initial concentration of zero.

these often operate on molecules upstream of the cascade represented here [56]. To keep the model simple, we incorporate the effects of feedback with a generic mechanism assuming that the feedback operates on the input molecule MAP4K, and that the feedback occurs due to a phosphorylation event that either activates or inactivates the kinase.

To incorporate feedback, the core model of the MAPK cascade is rewired so that the input MAP4K and output MAPKPP are connected through a feedback module (Fig 6A). Because feedback is through a phosphorylation event, we model feedback by reusing the phosphorylation cycle module in Fig 4C. The new model contains the MAP4K-I species, representing the inactive form of MAP4K. Note that in Fig 6A, the feedback ports that the input and output species connect to (referred to as X1 and X2) determine whether positive or negative feedback results. In the case of positive feedback, the active form of MAP4K is the phosphorylated form, i.e. X1 = XP and X2 = X (Fig 6B). For negative feedback, the active form of MAP4K is the unphosphorylated form, i.e. X1 = X and X2 = XP (Fig 6C).

Simulations of the MAPK cascade with feedback to steady-state are shown in Fig 6D, along with the corresponding simulation of the model without feedback for comparison. As has been predicted previously, negative feedback reduced ultrasensitivity [57, 58]. When viewed as an amplifier, negative feedback can increase the range of usable input concentrations at the expense of reduced gain [22, 59, 60]. The model with positive feedback exhibited bistability, a property seen in previous models of the MAPK cascade [56]. When the model was initialised in an active state, the response curve was virtually identical to the system without feedback. However, when the model was initialised to an inactive state, the response curve remained inactive for a wide range of input concentrations and only activated at high input concentrations.

## Benchmarking rate laws in a model of glycolysis

Kinetic models of metabolic systems make use of numerous rate laws, such as mass action, Michaelis-Menten and Hill equations. However, in many cases, these rate laws are not thermodynamically consistent. The bond graph approach builds on existing work by using thermodynamically independent parameters to ensure that rate laws are thermodynamically consistent [15, 44, 48]. In this section, we use glycolysis as an example to demonstrate the ability of bond graphs to swap out rate laws for one another and to benchmark the performance of alternative rate laws.

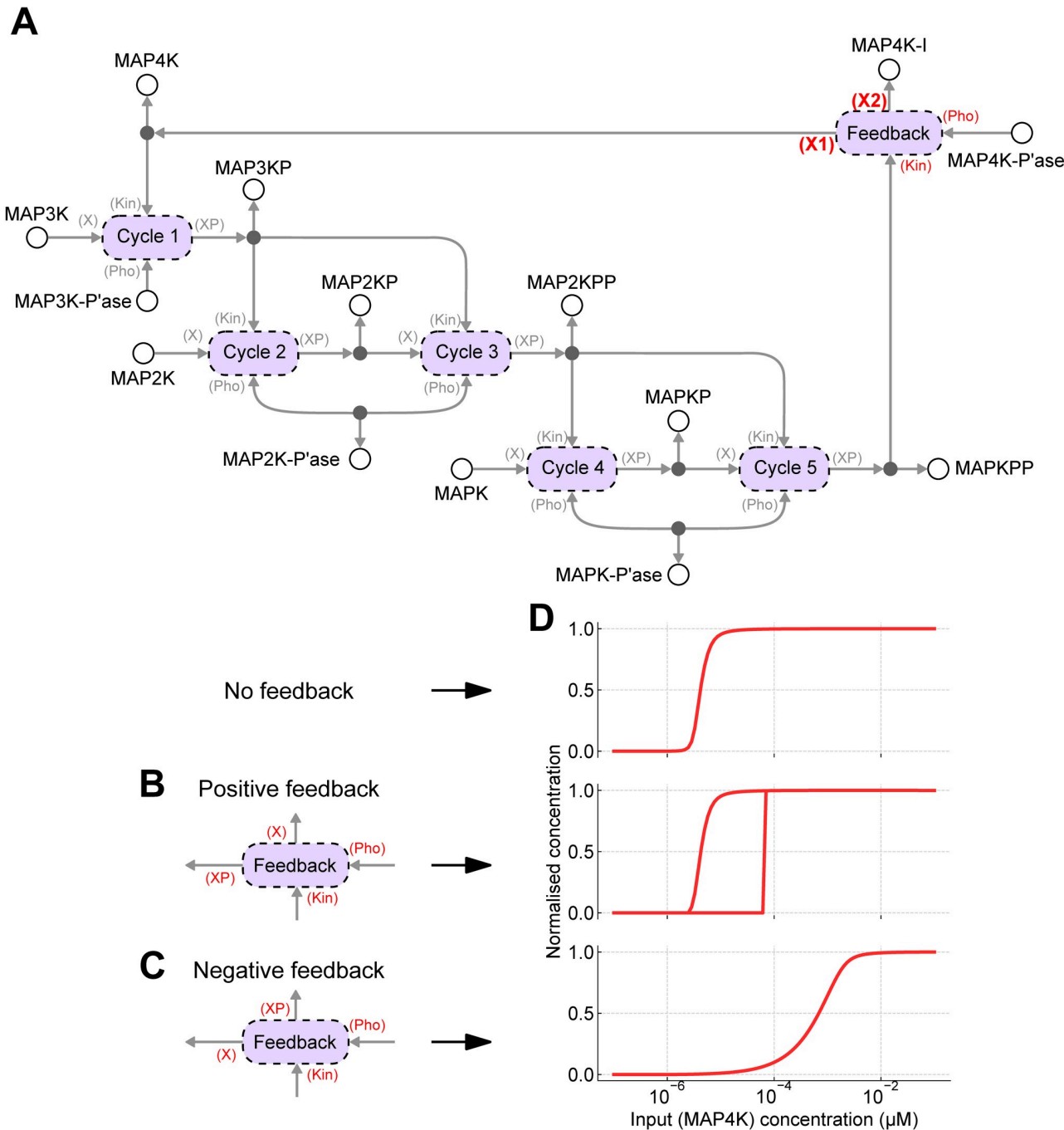

**Fig 6. Feedback within the MAPK cascade. (A)** Feedback can be added by modifying the core model of the MAPK cascade such that the input (MAP4K) and output (MAPKPP) are connected through a feedback module, which is implemented using the phosphorylation cycle defined in Fig 4C. The nature of the connections can give rise to both positive and negative feedback loops. ATP, ADP and Pi have been omitted for clarity. **(B)** For positive feedback, the active form of MAP4K is the phosphorylated species; **(C)** whereas for negative feedback, the active form of MAP4K is the dephosphorylated species. **(D)** The steady-state response (MAPKPP) of each system in response to changing input. (top) No feedback; (middle) positive feedback; (bottom) negative feedback. The initial conditions are the same as in Fig 5. The model with positive feedback is bistable, and the upper curve is obtained by setting the initial conditions for MAP3K, MAP2K and MAPK to zero, and instead using $x_{\mathrm{MAP3KP}} = 3$ nM, $x_{\mathrm{MAP2KPP}} = 1.2\ \mu$M, $x_{\mathrm{MAPKPP}} = 1.2\ \mu$M.

One can create a model of glycolysis by using the stoichiometry to define a high-level reaction structure with swappable modules for each enzyme, and then choose an appropriate rate law for each enzyme, depending on the fit to data (Fig 7). Indeed, this approach was taken by Gawthrop et al. [48].

A detailed rate law, used by Mason and Covert [5], is given by the equation

$$v = \bar{\kappa} e_0 \frac{e^{A^f/RT} - e^{A^r/RT}}{-1 + \prod_{s \in \mathcal{S}} \left(1 + \frac{e^{\mu_s/RT}}{R_{b,s}}\right) + \prod_{p \in \mathcal{P}} \left(1 + \frac{e^{\mu_p/RT}}{R_{b,p}}\right)} \tag{14}$$

where $\bar{\kappa}$ is a rate parameter, $\mathcal{S}$ is the set of all reactants, $\mathcal{P}$ is the set of all products and $R_{b,z}$ is the binding parameter associated with the substrate $z$. In the case of multiple stoichiometries, each binding site has a separate parameter $R_{b,z}$; $\mathcal{S}$ and $\mathcal{P}$ include multiple instances of such species (distinguished by numerical indices) in this scenario. For reactions where all reactants and products have a stoichiometry of one, the rate law is identical to convenience kinetics [44]. However, when multiple stoichiometries are involved (for example, in the enzyme *pps*, Fig 7A), this contains additional parameters. For this reason, we will refer to this rate law as the "generalised kinetics" (GK) rate law.

In many cases, it can be helpful to substitute complex rate laws with simpler ones, for example, to ease parameter estimation or to make mathematical analyses more tractable [61]. Thus, we constructed simplified versions of the generalised kinetics model (with parameters taken from Mason and Covert [5]) using both Michaelis-Menten kinetics (Eq 13) and mass action kinetics (Eq 6). In brief, the parameters were chosen to match the steady state of the generalised kinetics model. In the case of Michaelis-Menten kinetics, the binding parameters were chosen to match the behaviour of the enzyme to internal species where possible. Details of how parameters were derived for the simplified models are given in Appendix A of S1 Text, with parameters in S2 Table.

In order to obtain nonzero steady-state flows through the system, we assume that certain species (G6P, PYR, NAD, NADH, ATP, ADP, AMP, Pi, H, H2O) have a zero rate of change, modelling their replenishment through the environment. These are the external species, or "chemostats" in bond graph terminology [47].

**Transient perturbations.** We firstly tested each of the models by perturbing the concentrations of each of the internal species, causing transient shifts away from the reference steady state (Fig 8). The responses of the simplified models were firstly compared against the original generalised kinetics model by calculating the response time, which was defined as the time required for the system to return to within 5% of its maximum deviation. Distance from the reference steady state was calculated using the Euclidean norm

$$d = \sqrt{\sum_{s \in \mathcal{S}_i} (x_s - x_{s,ss})^2} \tag{15}$$

where $\mathcal{S}_i = \{\text{F6P}, \text{F16P}, \text{DHAP}, \text{GAP}, \text{13DPG}, \text{3PG}, \text{2PG}, \text{PEP}\}$ is the set of all internal species and $x_{s,ss}$ is the concentration of $s$ at the reference steady state. The response times are plotted in the top row of Fig 8. We also compared the models by plotting the concentration of PEP, the most downstream internal species against time (Fig 8; bottom row).

As seen in all perturbations, the Michaelis-Menten model matched with the original generalised kinetics model extremely well. This was expected as the Michaelis-Menten model was parameterised to match the behaviour of the generalised kinetics models with respect to the internal species. The minor differences between the models stem from the *fba* enzyme, which

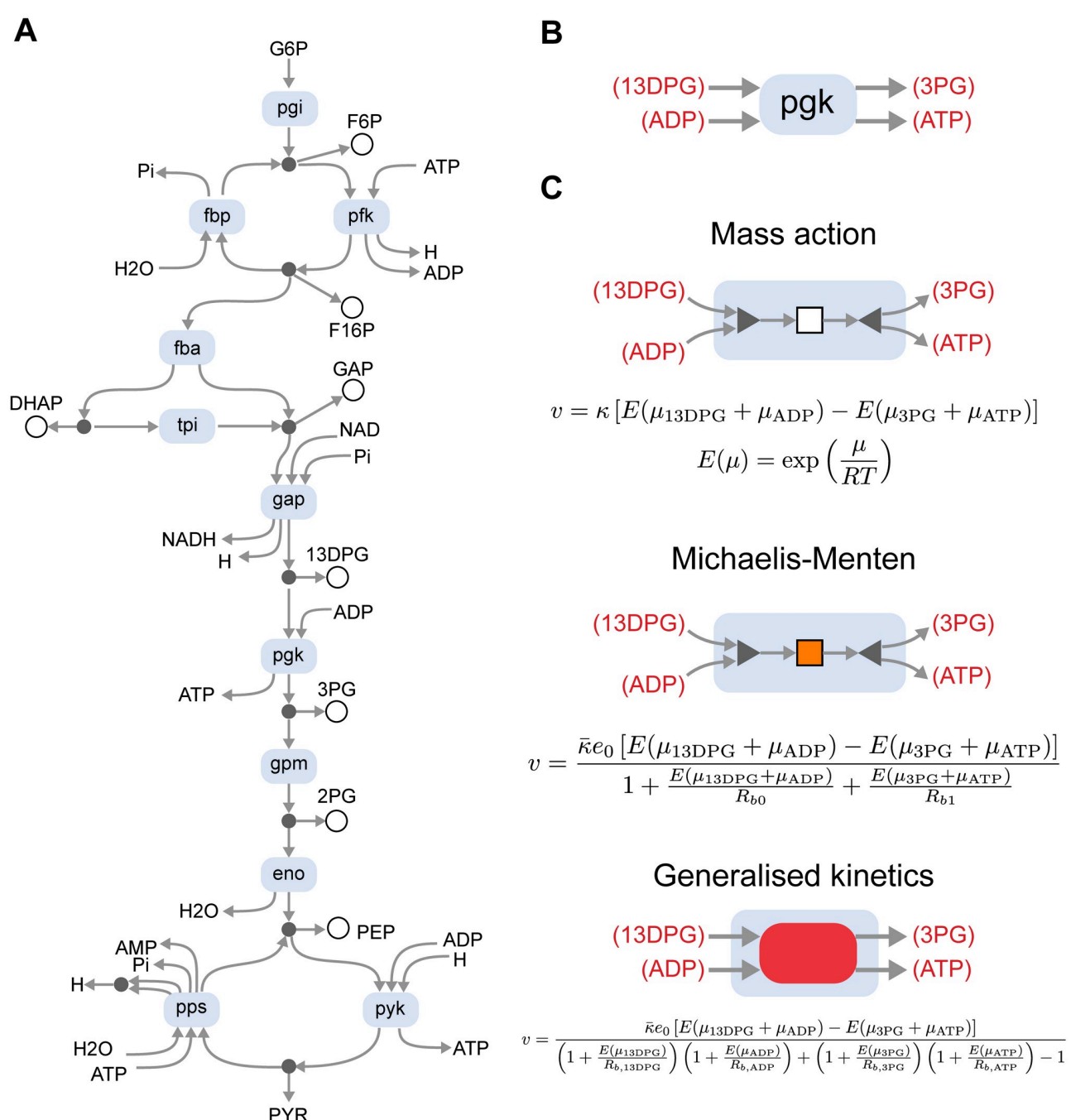

**Fig 7. Modelling of enzymes within the glycolysis pathway. (A)** A network-level representation of the system, where the blue modules are free to be swapped depending on the rate law. Species without circles are considered to be external to the system, and in cases where they occur more than once, they are connected by equal potential components (●, omitted for compactness) to ensure mass conservation. **(B)** For illustrative purposes, we show the rate laws for the pgk enzyme. **(C)** The enzyme can be modelled using the mass action (top), Michaelis-Menten (middle) or generalised kinetics (bottom) rate laws. The notation for the mass action and Michaelis-Menten components are defined in Fig 3B. Note that since generalised kinetics rate laws depend on the chemical potentials of all substrates (and not just their sums), they cannot be decomposed into smaller modules.

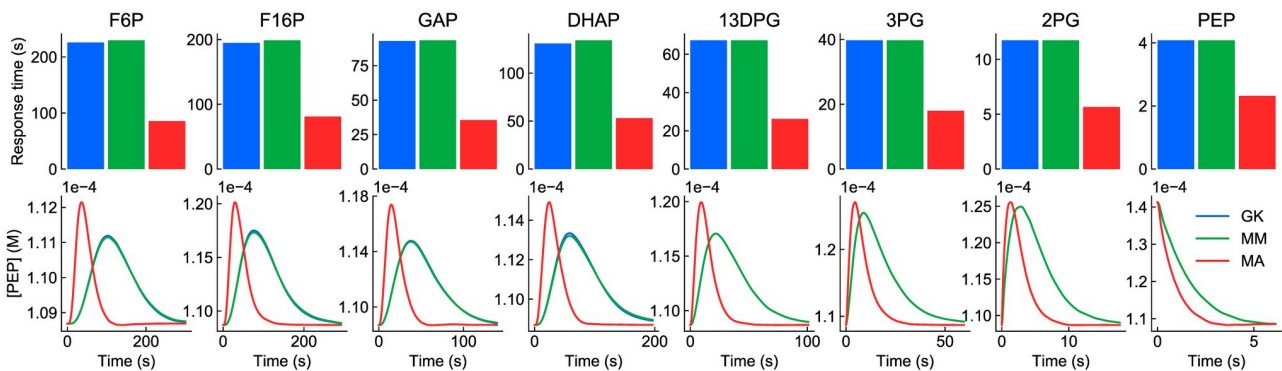

**Fig 8. Response of glycolysis models to perturbations to internal species.** Each of the species (column titles) had its concentration instantaneously increased by 30% from steady state. Top row: response times; bottom row: change in [PEP] over time. The colour key is blue: generalised kinetics (GK), green: Michaelis-Menten (MM), red: mass action (MA). In cases where the curve for the generalised kinetics model is not visible, it matches with the Michaelis-Menten model.

has two products and therefore could not be exactly matched to the generalised kinetics model (see Appendix A of S1 Text).

The mass action model behaved substantially differently from the generalised kinetics model with respect to the perturbations, notably having a faster response time. Additionally, the concentration of PEP had a greater maximum deviation from its steady-state value in response to perturbations of the more upstream species. Unlike the other rate laws, the law of mass action does not account for the rate-limiting step of enzyme complexes releasing product. Thus, these observations could potentially be attributed to the lack of saturation in the mass action rate law, causing increased reaction fluxes.

**Steady-state perturbations.** We also tested the response of the models to prolonged perturbations with external species, which caused the models to move to different steady states (Fig 9). Once again, the models were compared using the response time (top row) and concentration of PEP (bottom row). While all models have the same steady state without the

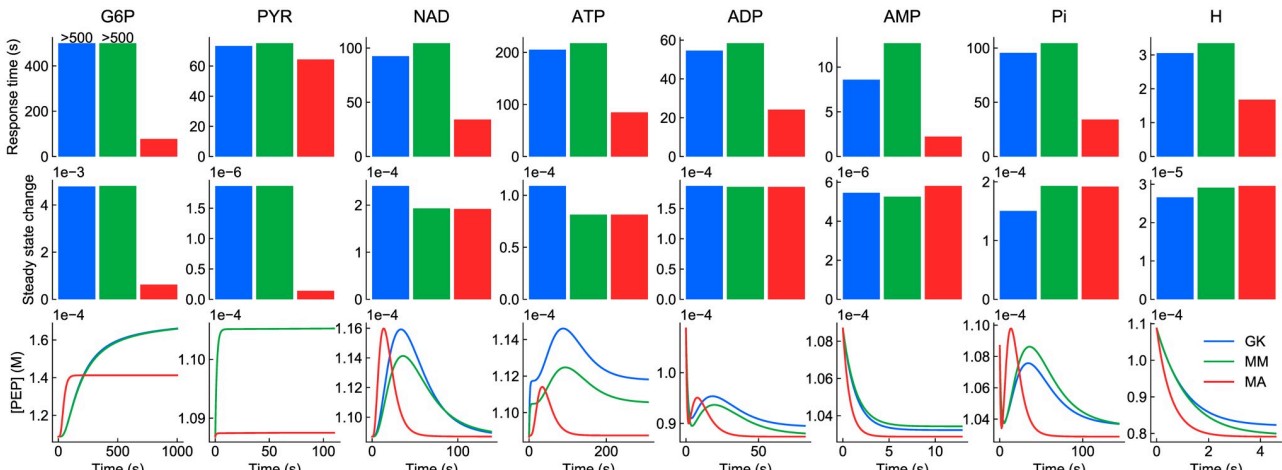

**Fig 9. Response of glycolysis models to prolonged perturbations to external species.** Each of the species (column titles) had its concentration instantaneously increased by 30% from steady state. Top row: response times; middle row: steady state deviation; bottom row: change in [PEP] over time. The colour key is blue: generalised kinetics (GK), green: Michaelis-Menten (MM), red: mass action (MA). The response of the model to NADH was omitted as there was a negligible change in steady state.

perturbation, they settle to different steady states under prolonged perturbations due to differences in how the rate laws react to changes in the concentration of boundary species. Accordingly, we also quantified the magnitude of the shift in steady state using the Euclidean norm (middle row).

While the Michaelis-Menten model still qualitatively resembled the generalised kinetics model, differences started to emerge when external species were perturbed, as there were insufficient parameters to match the behaviour in response to external species. These differences appeared to be most significant for perturbations to NAD, ATP and Pi. In general, the response time for the Michaelis-Menten model was slightly longer than the full model.

Following the trend for internal perturbations, the mass action model behaved significantly differently from the original model. The mass action model had a shorter response time and reached a different steady state in many cases.

These results would appear to suggest that saturation is an important property to consider when modelling the dynamic behaviour of metabolic networks, mirroring results from previous studies [62, 63]. Comparisons between the generalised kinetics and Michaelis-Menten models illustrate that while quantitative differences arise from simplifying out the complex binding properties of enzymes, simpler models may nonetheless be useful in studying the qualitative behaviour of metabolic networks, particularly under conditions where appropriate assumptions are satisfied.

**Energetics of the glycolysis pathway.** In addition to exploring fluxes and concentrations, bond graph models can be used to study the energetics of metabolic pathways, allowing modellers to incorporate thermodynamic measurements into methodologies for model parameterisation and validation. In some cases, analysing the transduction and dissipation of energy can result in novel insights and predictions [55].

The glycolysis pathway contains two points (*fba*/*fbp* and *pyk*/*pps*) at which carbon species are cycled by two enzymes while dissipating energy. These futile cycles (or "Cyclic Flow Modulation" [64]) are critical points of control, allowing the system to switch between glycolysis and gluconeogenesis [65, 66]. Much of this regulation is performed by allosteric regulation, which is not accounted for in this model. Nonetheless, the enzyme concentrations $e_0$ can be changed to model the effects of allosteric regulation.

We analyse the energetics of the generalised kinetics model of glycolysis in this section. To simplify our analysis, we switch off the *fbp* and *pps* enzymes (generally associated with gluconeogenesis) by setting $e_0$ to zero. The remaining reactions form a pathway, which we analyse at steady state. Using the methods of Gawthrop and Crampin [67], the glycolysis pathway can be defined as the sum of reactions

$$\text{pgi} + \text{pfk} + \text{fba} + \text{tpi} + 2\text{gap} + 2\text{pgk} + 2\text{gpm} + 2\text{eno} + 2\text{pyk}. \tag{16}$$

This pathway has the overall reaction

$$\text{G6P} + 3\text{ADP} + 2\text{NAD} + 2\text{Pi} \rightleftharpoons 2\text{PYR} + 3\text{ATP} + \text{H} + 2\text{NADH} + 2\text{H2O}. \tag{17}$$

Thus we can calculate the overall affinity of the pathway to be

$$
\begin{aligned}
A_{\text{glycolysis}} \quad &= \mu_{\text{G6P}} + 3\mu_{\text{ADP}} + 2\mu_{\text{NAD}} + 2\mu_{\text{Pi}} \\
&\quad - 2\mu_{\text{PYR}} - 3\mu_{\text{ATP}} - \mu_{\text{H}} - 2\mu_{\text{NADH}} - 2\mu_{\text{H2O}}
\end{aligned}
\tag{18a}
$$

$$
= A_{\text{pgi}} + A_{\text{pfk}} + A_{\text{fba}} + A_{\text{tpi}} + 2A_{\text{gap}} + 2A_{\text{pgk}} + 2A_{\text{gpm}} + 2A_{\text{eno}} + 2A_{\text{pyk}}. \tag{18b}
$$

**Table 1. Distribution of free energy changes in the glycolysis pathway.** The total free energy corresponds to the overall reaction G6P + 3ADP + 2NAD + 2Pi ⇌ 2PYR + 3ATP + H + 2NADH + 2H2O.

| Reaction | Affinity (kJ/mol) |
|---:|:---:|
| pgi | 43.4 |
| pfk | 81.0 |
| fba | 14.6 |
| tpi | 8.4 |
| gap | 51.5 (×2) |
| pgk | 23.9 (×2) |
| gpm | 13.5 (×2) |
| eno | 45.1 (×2) |
| pyk | 17.0 (×2) |
| Total | 449.4 |

The energy-based approach reveals a more detailed picture of how energy is dissipated throughout the pathway. Using the concentrations of each metabolite at steady state, one can calculate the affinity of each individual reaction. As expected, when scaled by the contribution of each reaction to the pathway, the affinities of the reactions add up to the total affinity (Table 1).

The total pathway affinity predicted by the model is higher than the experimentally measured values [68]. Furthermore, all reactions contribute significantly to the overall affinity, which differs from experimental measurements finding that the *pfk* and *pyk* are the predominant contributors to overall affinity, with the other reactions near equilibrium [68]. We note that for this particular model, the parameters were derived in the absence of standard free energies of formation [5]. Thus, a natural improvement to the model would be to use these values to parameterise models [69], which would likely improve the fit to experimental data. After incorporating data on thermodynamic constants, this model could be regarded as a validated module available for coupling with other reactions to form more comprehensive models. For example, recycling reactions such as the adenylate kinase and alcohol dehydrogenases could be added to study variations in the concentrations of ATP, ADP, AMP, NAD and NADH.

## Discussion

It is widely accepted that a modular approach is essential to developing large-scale models in systems biology. While significant progress has been made in using computational resources to support modular modelling, it remains challenging to ensure the integrated models are consistent with the laws of physics. In this paper, we have illustrated that bond graphs are both modular and physically consistent, allowing them to unify developments from both software and thermodynamical modelling. These principles were demonstrated by constructing physically consistent models of two well-studied systems using the modular properties of bond graphs.

To construct the model of the MAPK cascade, we took advantage of the ability of bond graphs to embed modules into reusable templates. Due to the presence of repeating motifs, the development of the model was substantially simplified, illustrating how similar concepts may be usable in streamlining the development of models of more complex systems. As demonstrated in Appendix B of S1 Text, this approach can be extended to use white-box modules with flexible interfaces. Furthermore, the merging of models can be automated through the

use of semantic annotations, and bond graphs have shown great potential in this space due to their biophysical detail [70].

In addition to embedding multiple components into a module, bond graphs also enable reactions to be modelled by a wide array of thermodynamically consistent rate laws. Using a network-level representation of glycolysis as a template, we showed how parts of a bond graph can be substituted for rate laws of varying complexity. This enabled a principled approach for benchmarking and comparing models of glycolysis with different levels of complexity. While not explored in this paper, a strength of this approach is that different parts of a model can be represented at different levels of granularity. This modular approach would allow one to study subsystems of interest using highly detailed models while using more manageable coarse-grained representations for the rest of the model.

Due to their modular nature, bond graphs are compatible with existing approaches to coupling reaction networks, as implemented in the SBML hierarchical package and PySB [28, 30]. These approaches are similar in that component definitions can be defined by either outer modules (black-box modularity) or be merged with other components (white-box modularity). This means that bond graphs could potentially be built on top of existing software infrastructure for model annotation, composition and simulation [24, 25, 71]. However, the bond graph approach expands on existing approaches in a few ways. Firstly, bond graphs explicitly account for the transfer of power, ensuring that models are thermodynamically consistent. Secondly, unlike in typical kinetic models, species components contain their own independent equations and parameters, allowing conflicts between models to be resolved. Finally, the approach is generalisable to multi-physics systems such as mechanobiology and electrophysiology.

An ongoing challenge to creating truly large-scale bond graph models is increasing the number of models available to modellers. Ideally, one would look to convert existing kinetic models in the literature into bond graphs. However, while bond graphs are readily converted into kinetic models, the conversion in the other direction is not always possible as existing kinetic models often use irreversible reactions or fail to satisfy detailed balance [47, 72, 73]. Thus, to maintain the advantages of using bond graphs, an avenue of future work would be to automatically generate bond graph versions of existing biomodels, such as those in the BioModels repository [74] or the Physiome Model Repository [75], and where necessary correcting thermodynamic inconsistencies while minimally changing model behaviour. In cases where the energetics of a process are considered negligible (for example gating in ion channels and neural control in locomotion), one may choose to embed the kinetics of such processes inside control elements of a bond graph [76]. This allows a kinetic model to be directly added to a bond graph as a "black box" in which the energetics underlying its dynamic behaviour are ignored.

Because bond graphs are inherently a modular and declarative representation, they are well suited for taking advantage of developments in programmatic modelling where models are constructed through a series of instructions from the software [7, 30]. Indeed, the models in this paper were constructed using the BondGraphTools Python package [31], a highly flexible and automatable approach for model construction that mirrors the approach of existing packages used within the systems biology community [30, 71]. Embedding bond graphs within a programmatic environment enables the construction of models using higher-level descriptions, allowing modellers define models using a relatively parsimonious set of commands. Previous work has shown that bond graphs can be constructed from a stoichiometric matrix [48] and a series of reactions [31]. Nonetheless, further work is needed to help automate model construction, in particular making use of rule-based approaches for constructing models of highly complex interactions between proteins, ligands and receptors [77, 78].

While we used the Python package BondGraphTools for the benchmarking and simulation in this paper, it would also be relatively straightforward to use typical systems biology simulators (for example COPASI [25], Tellurium [71] or PySB [30]) after a change of parameters. However, we note that these programs do not allow species equations to be specified. Thus, we advocate for retaining a bond graph representation of the model as a core representation that can be later modified should the need arise. We are working on exporters to convert bond graphs into more commonly used formats such as SBML and CellML to aid model reproducibility. A longer-term goal is to define bond graphs in SBML and CellML. CellML describes the mathematics of physical systems, thus bond graphs are generally straightforwardly defined in this standard. However, to implement the approach in SBML, one would need to extend the standard so that species have their own equations and parameters, potentially through a separate package.

In writing this paper, we hope to motivate the uptake of network thermodynamic models (and bond graph models in particular) by the systems biology community. One of the objectives of the Physiome Project is to embed bond graphs within the CellML modelling language [79]. This will facilitate the development of more user-friendly tools, allowing the conversion of existing models into bond graphs and subsequently coupling them. As the bond graph methodology gradually becomes more widely adopted by the community, we anticipate that existing mathematical and computational methods in the systems biology space will be used in conjunction with the bond graph approach, allowing biomodels to be integrated in a physically consistent manner.

In conjunction with the programmatic approach, bond graphs provide a useful framework for updating models and recording their provenance. In the development of the model of the MAPK cascade, we showed that incremental changes could be made to incorporate feedback. Through a modular approach, these changes were made by changing the components and connections within the network rather than deriving new equations entirely. Furthermore, these incremental changes can be recorded within the code used to construct such models, which could potentially enable an automated framework for profiling model provenance in the future.

In order to develop comprehensive models of cells, ongoing and future work will focus on expanding the range of cellular processes that bond graphs can represent. While significant progress has been made in metabolic modelling and transport processes, how to model gene regulation and signalling in an energy-based framework remains an open question. Modelling such processes will likely require theoretical groundwork to be established for:

1. modelling discrete and stochastic systems

2. choosing an appropriate level of model granularity to model each biological process

3. dealing with the properties of macromolecules in a modular manner, so that the number of associated parameters remains manageable

While the Network Thermodynamics approach implemented using bond graphs requires modellers to take more care in developing their models in the short term, we believe that taking this approach will make large-scale models more robust, reusable and ultimately more useful to the systems biology community in the years to come.

## Supporting information

**S1 Text. Supplementary material. Appendix A**: Details of parameter identification for the MAPK cascade and glycolysis. **Appendix B**: Outline of a white-box approach to modelling the

MAPK cascade. **Appendix C**: Tutorial on basic BondGraphTools functionality. **Fig A**: Comparison of the bond graph model of the MAPK cascade to the Huang and Ferrell model. **Fig B**: A white-box approach to defining a model of the MAPK cascade. **Fig C**: A bond graph model of the reaction $A + B \rightleftharpoons C$. **Fig D**: BondGraphTools rendering of the reaction $A + B \rightleftharpoons C$. **Fig E**: Simulation of the reaction $A + B \rightleftharpoons C$.
(PDF)

**S1 Table. Parameters for the MAPK cascade.**
(XLSX)

**S2 Table. Parameters for the glycolysis model.**
(XLSX)

## Author Contributions

**Conceptualization:** Michael Pan, Joseph Cursons, Edmund J. Crampin.

**Data curation:** Michael Pan.

**Formal analysis:** Michael Pan, Peter J. Gawthrop.

**Funding acquisition:** Edmund J. Crampin.

**Investigation:** Michael Pan.

**Methodology:** Michael Pan, Peter J. Gawthrop, Edmund J. Crampin.

**Project administration:** Michael Pan, Edmund J. Crampin.

**Software:** Michael Pan.

**Supervision:** Edmund J. Crampin.

**Validation:** Michael Pan.

**Visualization:** Michael Pan, Peter J. Gawthrop, Joseph Cursons.

**Writing – original draft:** Michael Pan.

**Writing – review & editing:** Michael Pan, Peter J. Gawthrop, Joseph Cursons, Edmund J. Crampin.

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
