## [Decision Letter · Decision Letter 0]

25 Aug 2021

Dear Dr. Pan,

Thank you very much for submitting your manuscript "Modular assembly of dynamic models in systems biology" for consideration at PLOS Computational Biology. As with all papers reviewed by the journal, your manuscript was reviewed by members of the editorial board and by several independent reviewers. The reviewers appreciated the attention to an important topic. Based on the reviews, we are likely to accept this manuscript for publication, providing that you modify the manuscript according to the review recommendations.

Sincerely,

Anders Wallqvist

Associate Editor

PLOS Computational Biology

Daniel Beard

Deputy Editor

PLOS Computational Biology

[LINK]

Reviewer's Responses to Questions

**Comments to the Authors:**

Reviewer #1: This is a manuscript that essentially summarizes the bond graph approach of network thermodynamics and proposes it to be a suitable method to modularize the development of large-scale models (such as whole-cell models). Overall I find the manuscript to be very clear and accessible and overall I think this will interest the readership of the journal.

My only major concern regards the glycolysis example used, where they chose to keep AMP, ADP, ATP, NAD and NADH constant. These conserved species are recycled and are too important to ignore by keeping them constant. It would have been much more interesting if they had included their recycling reactions (like alcohol dehydrogenase for NAD/NADH and an ATPase + an adenilate kinase for ATP/ADP/AMP. This would be much more in line with what they did in the MAPK example. However, I do understand that this is not essential for the purpose of their argument (the utility of bond graph approach), but it would be a whole lot more interesting. Importantly, doing that would allow a better energetic analysis of glycolysis...

There are a number of minor points that nevertheless I think they could easily address and which would make the manuscript much more useful for readers:

1- Thermodynamics

To be fair to readers it is important to be accurate in citing prior work. In this regard I bring attention to your sentence in the Introduction:

"While the whole-cell modelling community has emphasised the need to use physically measurable parameters, it is only recently that the importance of thermodynamics in these models has been acknowledged [5]."

the importance of thermodynamics in this type of models has been invoked prior to this by several authors, such as your references [32] and [33], Henry et al (2006), Soh et al (2012), Lubitz et al (2010) and very much in this large-scale kinetic model context by Stanford et al (2013) (disclaimer: I am a co-author of the latter and I am not writing this seeking a citation; rather I feel that a reader should not be mislead that your ref. [5] was the first time thermodynamics had been thought of in this context).

2- Functional modularity: I thought it could be pertinent to mention the work of Del Vecchio et al (2008) in this context.

3- Figure 2 is very useful, but it would be better to make 2B actually use the SBGN standard. I note that it would require only one change to make it compliant: the arrows should not point into the reaction nodes, only into the species nodes. Then you couldsay that this representation follows the standard (rather than it is similar to).

4- The MAPK are not only significant clinically for being involved in many cancers, but also in inflammation (P38 pathway).

5- Fig. 4, I note that in panel B you represent the phosphatase in the direction which seems to be the inverse of the biological flow. Normally phosphatases go from XP -> X rather than X -> XP (thermodynamically this direction is very unfavorable unless in the presence of a massive and unphysiological concentration of Pi... It would be much better if you would represent 4B in the physiological direction.

References that I cited which are not in the manuscript:

- Henry CS, Jankowski MD, Broadbelt LJ, Hatzimanikatis V. (2006) Genome-scale thermodynamic analysis of Escherichia coli metabolism. Biophys J. 90(4):1453-61.

- Soh KC, Miskovic L, Hatzimanikatis V. (2012) From network models to network responses: integration of thermodynamic and kinetic properties of yeast genome-scale metabolic networks. FEMS Yeast Res. 12(2):129-43.

- Lubitz T, Schulz M, Klipp E, Liebermeister W (2010) Parameter balancing for kinetic models of cell metabolism. J Phys Chem B 114: 16298-16303.

- Stanford NJ, Lubitz T, Smallbone K, Klipp E, Mendes P, Liebermeister W (2013) Systematic construction of kinetic models from genome-scale metabolic networks PLoS ONE 8:e79195

- Del Vecchio D,Ninfa AJ, Sontag ED (2008) Modular cell biology: retroactivity and insulation. Mol Syst Biol 4:161.

Reviewer #2: The manuscript by Michael Pan et al describes a physics-based approach for modularity in biological modeling. They describe modular construction of models based on bond graphs.

The first part of the manuscript described bond graphs, defining reaction network (RN) through thermodynamic (TD) notations, expressing all kinetic rate constants through TD parameters. Mathematically it means introducing a larger number of parameters that are energy-based and independent, contrary to dependent parameters in a reaction network. Visually it corresponds to decomposing the RN (usually represented as a bipartite graph of species and reaction nodes) into 4-partite graph (separate nodes for species, reactions, mass-conservation check-points and energy conservation check-points). This part is clearly written and very useful.

The second part describes using bond graphs to construct and combine modules into larger biomedical models. This is a nice and comprehensive description of modular approach, but the authors have not compared their approach with SBML hierarchical model composition package. It is described at http://co.mbine.org/standards/sbml/level-3/version-1/comp. Although SBML hierarchical package is too technical and not written for a general audience, it introduces the same concepts of black-box versus white-box encapsulation and “ports” as interfaces between a module and its containing model. A modular SBML-based approach is implemented in iBioSim (https://async.ece.utah.edu/tools/ibiosim/). Thus, I would appreciate if authors could address the following comments:

1. It would be useful to have more input on what in Figures 4, 6 and 7 is different from a regular modular composition of RNs using ports.

2. When reading manuscript beyond page 13, energy conservation is not mentioned anymore and bond graphs look as RN defined with new TD parameters. Is it true?

3. Is all TD machinery internal for each module and not exposed? Can we “mix and match” by defining some modules in terms of TD and some in terms of RN?

4. Can benchmarking rate laws and effects of perturbations be computed in a regular RN simulator (COPASI, Tellurium, PySB, etc) after a simple change of parameters?

5. Compare their modular approach with SBML hierarchical composition.

6. Discuss whether bond graphs can be described as an extension to SBML standard.

Reviewer #3: This is a nicely written paper on the use of bond-graphs to build composable systems biology models. I only have a couple of minor points to make and for the authors to clarify in the paper:

1. I assume the authors know that SBML supports a white/black box approach to composition in the hierarchical package:

https://pubmed.ncbi.nlm.nih.gov/26528566/

This should be cited, perhaps on page 6

2. One important question is whether bond-graph representations (I think they are stored as json files?) could be converted to SBML given the ubiquity of simulators for this format. One could in fact imagine the bondgraph description being a higher level of description that can be converted to a simpler SBML file where the bond graph representation is the model that is edited and improved. The SBML this acts as an intermediary. Or do the authors envisage adding new annotations to SBML (or a package) to extend SBML to store bondgraph models?

3. One the major disadvantages of the bondgraph approach is its verbosity (plus the terminology is not as well-known as the more simple kinetic formalisms). The same problem arises with Vivarium but this tool has also made the fatal mistake of mixing specification with implementation. Although perhaps not applicable to this manuscript, software tooling will be a major impediment to the widespread use of bond graphs. I looked at bondgraphstools and there might be a case of adding a higher layer of abstraction because currently is seems quite laborious to build even small models, I also wonder whether in its current state it could scale to whole-cell models (the same applies to vivarium) although the composition feature would help. There might need to be additional abstraction levels.

3. Page 10, the authors mention K as the thermodynamic parameter, what is that exactly? I see that they cite the appendix for further info but a mention in the main text as to what K is would be useful.

4. Page 21, second line where they site ref 15 with respect to linearization due to feedback. I believe this was first mention by Sauro and Ingalls (https://arxiv.org/abs/0710.5195, MAPK Cascades as Feedback Amplifiers) and has since been brought up since by other authors.

5. Page 25 second paragraph it says that the mass-action model reached a different steady state. I don’t think that can be true because at steady state the mass-action and Michalis formalisms as exactly the same. It must be that the bondgraph model is someone different. Differences in time scales I can understand but not differences in the steady state.

**Have the authors made all data and (if applicable) computational code underlying the findings in their manuscript fully available?**

Reviewer #1: Yes

Reviewer #2: Yes

Reviewer #3: Yes

PLOS authors have the option to publish the peer review history of their article (what does this mean?). If published, this will include your full peer review and any attached files.

Reviewer #1: **Yes: **Pedro Mendes

Reviewer #2: No

Reviewer #3: No

Figure Files:

Data Requirements:

Reproducibility:

References:

---

## [Editor Report · Decision Letter 1]

30 Sep 2021

Dear Dr. Pan,

We are pleased to inform you that your manuscript 'Modular assembly of dynamic models in systems biology' has been provisionally accepted for publication in PLOS Computational Biology.

Best regards,

Anders Wallqvist

Associate Editor

PLOS Computational Biology

Daniel Beard

Deputy Editor

PLOS Computational Biology

---

## [Editor Report · Acceptance letter]

10 Oct 2021

PCOMPBIOL-D-21-01387R1 

Modular assembly of dynamic models in systems biology

Dear Dr Pan,

I am pleased to inform you that your manuscript has been formally accepted for publication in PLOS Computational Biology. Your manuscript is now with our production department and you will be notified of the publication date in due course.

With kind regards,

Zsofia Freund
